# Crab-Eating Monkey Acidic Chitinase (CHIA) Efficiently Degrades Chitin and Chitosan under Acidic and High-Temperature Conditions

**DOI:** 10.3390/molecules27020409

**Published:** 2022-01-09

**Authors:** Maiko Uehara, Chinatsu Takasaki, Satoshi Wakita, Yasusato Sugahara, Eri Tabata, Vaclav Matoska, Peter O. Bauer, Fumitaka Oyama

**Affiliations:** 1Department of Chemistry and Life Science, Kogakuin University, Tokyo 192-0015, Japan; bd19001@ns.kogakuin.ac.jp (M.U.); bm19024@g.kogakuin.jp (C.T.); wsat.chorus.618@gmail.com (S.W.); bt79310@ns.kogakuin.ac.jp (Y.S.); tbt16024@yahoo.co.jp (E.T.); 2Japan Society for the Promotion of Science (PD), Tokyo 102-0083, Japan; 3Laboratory of Molecular Diagnostics, Department of Clinical Biochemistry, Hematology and Immunology, Homolka Hospital, Roentgenova 37/2, 150 00 Prague, Czech Republic; vaclav.matoska@homolka.cz (V.M.); peter.bauer@bioinova.cz (P.O.B.); 4Bioinova JSC, Videnska 1083, 142 20 Prague, Czech Republic

**Keywords:** acidic chitinase, chitin, chitosan, chitooligosaccharides, FACE method

## Abstract

Chitooligosaccharides, the degradation products of chitin and chitosan, possess anti-bacterial, anti-tumor, and anti-inflammatory activities. The enzymatic production of chitooligosaccharides may increase the interest in their potential biomedical or agricultural usability in terms of the safety and simplicity of the manufacturing process. Crab-eating monkey acidic chitinase (CHIA) is an enzyme with robust activity in various environments. Here, we report the efficient degradation of chitin and chitosan by monkey CHIA under acidic and high-temperature conditions. Monkey CHIA hydrolyzed α-chitin at 50 °C, producing *N*-acetyl-d-glucosamine (GlcNAc) dimers more efficiently than at 37 °C. Moreover, the degradation rate increased with a longer incubation time (up to 72 h) without the inactivation of the enzyme. Five substrates (α-chitin, colloidal chitin, P-chitin, block-type, and random-type chitosan substrates) were exposed to monkey CHIS at pH 2.0 or pH 5.0 at 50 °C. P-chitin and random-type chitosan appeared to be the best sources of GlcNAc dimers and broad-scale chitooligosaccharides, respectively. In addition, the pattern of the products from the block-type chitosan was different between pH conditions (pH 2.0 and pH 5.0). Thus, monkey CHIA can degrade chitin and chitosan efficiently without inactivation under high-temperature or low pH conditions. Our results show that certain chitooligosaccharides are enriched by using different substrates under different conditions. Therefore, the reaction conditions can be adjusted to obtain desired oligomers. Crab-eating monkey CHIA can potentially become an efficient tool in producing chitooligosaccharide sets for agricultural and biomedical purposes.

## 1. Introduction

Chitin is a β-1,4-linked polymer of the *N*-acetyl-d-glucosamine (GlcNAc) units [1,2]. It is an integral component of the exoskeletons of crustaceans and insects, the microfilarial sheaths of parasitic nematodes and fungal cell walls [1,2,3]. Thus, this polysaccharide is the second most abundant polysaccharide in nature.

Chitosan is a linear amino polysaccharide composed of d-glucosamine (GlcN) and GlcNAc units. It is a heterogeneously or homogeneously deacetylated derivative of chitin [4], forming “block-type” or “random-type” chitosan, respectively [5,6].

Chitinases are enzymes that hydrolyze the chitin polymers. Although mammals do not produce chitin, mice and humans synthesize two active chitinases: chitotriosidase (CHIT1) and acidic chitinase (CHIA) [2,3,7]. CHIA has been identified as a compensatory enzyme for CHIT1 [8,9]. Based on sequence similarities, CHIT1 and CHIA belong to the Glycoside Hydrolase Family 18 (GH18) [8,10,11,12,13]. The conserved sequence in GH18 involved in catalysis is DXXDXDXE, where E is assumed to be the catalytic residue [11]. Both CHIT1 and CHIA digest natural chitin and chitosan, possibly through endo-chitinase activity [7,8,14].

Previous studies have shown associations between CHIA expression and specific pathophysiological conditions. For example, CHIA expression is upregulated during allergic airway responses in mouse models of asthma [15,16]. Polymorphisms in the Chia gene are associated with bronchial asthma in humans [17,18,19]. Recent studies using CHIA-deficient mice have shown that CHIA is a constitutively expressed enzyme necessary for chitin degradation in the airways to maintain pulmonary functions [7,20]. In addition, Chia is a protease-resistant chitinase under gastrointestinal conditions [21,22,23,24].

Chitooligosaccharides are degradation products of chitin or chitosan. They possess remarkable anti-microbial, anti-tumor, and anti-inflammatory bioactivities and can be used in drug delivery systems [25,26,27,28,29]. Chitooligosaccharides also function as biostimulators in plant growth and are used as biopesticides and biofertilizers in agriculture [29,30]. The application of chitinase and acetyl-glucosaminidases for industrial purposes has been reported [31,32,33]. Generally, chitooligosaccharides are obtained by chemical degradation or enzymatic hydrolysis [34,35]. Enzymatic methods using mammalian sources can reduce the cost while increasing the quality of production due to their safety and the simplicity of the process control [36]. We have reported that Chia residues in porcine pepsin preparations exhibit chitinolytic activity and degrade chitosan under stomach conditions [37], and that mouse Chia effectively produces variable length chitooligosaccharides from random-type chitosan [38].

Crab-eating monkey (*Macaca fascicularis*) is one of the most crucial nonhuman primate animal models in biomedical research [39,40]. These primates feed on crabs and other chitin-containing organisms such as crustaceans and insects [41]. Previously, we performed a gene expression analysis and showed that the monkey expresses a high level of CHIA mRNA in the stomach [42]. Crab-eating monkey CHIA is 50 times more active than the human analogue [43]. Recently, we reported that crab-eating monkey CHIA had robust chitinolytic activity under a broad range of pH conditions and high thermal stability [44].

Here, we aimed to examine whether this enzyme could be useful for producing chitooligosaccharides from several types of chitin and chitosan substrates. We show that crab-eating monkey CHIA efficiently degrades the substrates while providing different sets of chitooligosaccharides under acidic and high-temperature conditions, and that the pattern and amount of such products can be regulated to achieve the enrichment of certain oligomers.

## 2. Results

### 2.1. Preparation of Recombinant Monkey CHIA

Crab-eating monkey CHIA (monkey CHIA) has robust chitinolytic activity with a stable performance at high temperatures and low pH [44]. Monkey CHIA maintains its activity under a broad range of pH (pH 1.0–7.0), with the maximum at pH 5.0 and the second peak at pH 2.0 [44]. Monkey CHIA has also been shown to function at high temperatures, demonstrating its thermostability with peak activities at pH 2.0 and 55 °C and at pH 5.0 and 70 °C [44]. Thus, we evaluated this enzyme as a potentially useful tool for the robust and controlled production of bioactive chitooligosaccharides under various conditions.

We expressed monkey CHIA as a fusion protein (Protein A-monkey CHIA-V5-His) using the pEZZ18 system in *E. coli* (Appendix A). We prepared the protein using the protocol reported previously [45]. The recombinant enzyme was analyzed by SDS-polyacrylamide gel electrophoresis (PAGE), followed by Western blotting using an anti-V5 antibody. We confirmed the protein as the major band of around 68 kDa (Appendix A). Next, the gel was stained with SYPRO Ruby (Appendix A), detecting additional bands not visible by Western blot. As reported previously [45], the bands around 55 kDa with chitinolytic activity are truncated forms of CHIA. The bands around 50 kDa were proteins unrelated to CHIA, with no chitinolytic activity. We obtained 840 µg of recombinant monkey CHIA from the 3 L culture of the transformed *E. coli*. The concentration of monkey CHIA used in this report was 0.7 µg/µL.

### 2.2. Monkey CHIA Efficiently Degrades Chitin Substrates at High Temperatures

To compare the chitin degradation under physiological and high temperatures, we incubated α-crystalline chitin (α-chitin) with monkey CHIA in McIlvaine’s buffer at pH 5.0 and 37 °C or 50 °C for 1, 5, and 24 h (Figure 1A and Appendix A). The substrate was mainly digested to (GlcNAc)_2_, with higher efficiency at 50 °C at each time point (Figure 1B and Appendix A).

Next, we examined the time-dependent accumulation of the degradation products at 50 °C after 1, 3, 5, 24, 48, and 72 h of incubation (Figure 1C and Appendix A). The levels of chitooligosaccharides grew during the whole incubation, indicating no inactivation of CHIA. The quantification of (GlcNAc)_2_ levels is shown in Figure 1D and Appendix A.

### 2.3. P-Chitin Is a Superior Substrate for GlcNAc Dimer Production

To evaluate monkey CHIA activity toward different substrates, we first incubated α-chitin, colloidal chitin (prepared from α-chitin), and P-chitin with the enzyme at pH 2.0 or pH 5.0 50 °C for 1, 3, 5, 24, 48, and 72 h.

We observed (GlcNAc)_2_ production from all used substrates at both pH levels (Figure 2 and Appendix A). The results indicate that the best substrate for (GlcNAc)_2_ production is P-chitin at either of the tested pH levels (Figure 2A and Appendix A). Low amounts of trimers were also obtained from this substrate. In the degradation of colloidal chitin under both pH conditions, (GlcNAc)_2_ levels produced by monkey CHIA were increasing with incubation time at pH 5.0 (Figure 2B and Appendix A). At pH 2.0, on the other hand, the accumulation plateaued after 48 h. As for α-chitin, dimer levels plateaued after a 24 h incubation at pH 2.0, while at a higher pH, the dimers gradually accumulated over the whole incubation period of 72 h (Figure 2C and Appendix A). Generally, monkey CHIA performed more efficiently at pH 5.0 with each substrate.

### 2.4. GlcNAc Dimer Production from Chitosan

Chitosan is a partially deacetylated derivative of chitin [4]. Previously, we showed the pattern of chitooligosaccharide production from mealworm larvae and fly wings under chicken, pig, and marmoset gastrointestinal-like conditions [41]. Recently, we reported that mouse Chia effectively degrades random-type chitosan oligomers of variable lengths under pH conditions mimicking stomach and lung tissues [38]. Published data suggest that chitosan substrates could be good sources for chitooligosaccharide production by monkey CHIA.

We incubated block-type or random-type chitosan with a similar deacetylation degree (D.D.; ~70%) with monkey CHIA at pH 2.0 or 5.0 and 50 °C for 1, 3, 5, 24, 48, and 72 h. Various chitooligosaccharides were obtained with higher efficiency at pH 2.0 (Figure 3 and Appendix A).

The degradation of the random-type chitosan was quite visible under both pH conditions. Incubation at pH 2.0 mainly provided (GlcNAc)_2_ and (GlcNAc)_3_ at pH 2.0, followed by (GlcNAc)_5_ and (GlcNAc)_6_ with a gradual accumulation, and a plateau from 24 h onwards. At pH 5.0, the presence of dimers was more prominent after short incubation times (1–5 h), with the continuous increase of all oligomers throughout the whole reaction time (Figure 3A and Appendix A).

Incubation of block-type chitosan with monkey CHIA resulted mainly in dimer and trimer production with relatively high levels of (GlcNAc)_6_ at pH 2.0. Under this condition, other oligomers occur after longer incubation (>24 h). Interestingly, (GlcNAc)_3_ appears to be the most abundant degradation product at pH 5.0, followed by (GlcNAc)_6_ and (GlcNAc)_2_ (Figure 3B and Appendix A).

The degradation products of random-type chitosan were produced in large amounts compared with block-type chitosan. Thus, monkey CHIA produced more chitooligosaccharides from random-type chitosan.

### 2.5. Pattern of the Chitooligosaccharides Produced by Monkey CHIA from Different Chitin and Chitosan Substrates

To compare the oligomer’s pattern obtained from various substrates, we incubated P-chitin, α-chitin, colloidal chitin, and block-type and random-type chitosan with monkey CHIA in McIlvaine’s buffer at pH 5.0 and 50 °C for 72 h and analyzed the degradation products.

Monkey CHIA produced the highest levels of (GlcNAc)_2_ from P-chitin, followed by α-chitin and colloidal chitin (Figure 4A,B, Appendix A, and Appendix A). As for chitosan, more efficient degradation occurred in random-type rather than block-type chitosan (Figure 4A,B, Appendix A, and Appendix A). The pattern of the produced chitooligosaccharides also differed between the chitosan types (Figure 4B and Appendix A). The size distribution of the produced chitooligosaccharides is shown in Figure 4C.

Except for the block-type chitosan, whose most abundant degradation product was (GlcNAc)_3_, all other substrates were hydrolyzed, mainly to (GlcNAc)_2_. Chitosan substrates tend to be degraded to oligomers with more diverse sizes.

## 3. Discussion

In this study, we show the optimal temperature (50 °C) and pH (pH 5.0) for the degradation of chitin and chitosan by crab-eating monkey CHIA (Figure 1). This enzyme’s thermo- and acid stability enable the substrates to process for prolonged periods (up to 72 h), resulting in a more varied set of chitooligosaccharides (Figure 2 and Figure 3).

The pattern and amount of the chitooligosaccharides obtained depended on the incubation time (Figure 3). Moreover, the pattern of the products obtained from block-type chitosan was also associated with the pH conditions (Figure 3B). We suggest that with the short-time incubation of chitosan under suitable pH conditions, the production of chitooligosaccharides with specific lengths can be achieved (Figure 3; 3 and 5 h incubation). It is clear that the GlcNAc dimer and longer chitooligosaccharides can be obtained in different patterns using various substrate types and pH conditions.

For a robust generation of GlcNAc dimers and different-sized chitooligosaccharides, P-chitin and random-type chitosan, respectively, represent the most suitable substrates (Figure 4). Modifying the reaction time, pH, temperature, and substrate, it is possible to at least partially control the chitin and chitosan degradation pattern and obtain the desired enrichment of certain oligomers.

As for the substrates, the degree of deacetylation influenced the amount of degradation products from chitosan and chitin substrates. We previously reported that the degradation products from β-chitin with higher D.D. were more abundant in comparison with α-chitin [37,38]. In addition, the degradation products differ significantly between block- and random-type chitosan due to the substrate specificity of mouse Chia [38]. The block-type chitosan has amorphous regions with highly deacetylated areas (GlcN-rich), while in the random-type chitosan, the GlcNAc and GlcN residues are interspersed. In this study, we show that monkey CHIA preferentially degrades the randomly placed GlcNAc-rich regions, producing more variable-sized chitooligosaccharides and influencing their amounts (Figure 4B and Appendix A).

Degradation of chitosan by monkey CHIA resulted in chitooligosaccharides migrating with the (GlcNAc)_1–6_ standard, as shown by the FACE method. In addition, minor bands were observed below the main bands. It has been shown that *Serratia marcescens* chitinases can degrade chitosan to hetero-chitooligomers consisting of GlcNAc and GlcN residues [46,47], and we already reported a similar observation with porcine Chia [37]. Therefore, the minor bands obtained by monkey CHIA may represent such hetero-chitooligosaccharides.

This study evaluated chitooligosaccharides produced from chitin and chitosan using FACE, because this method is very sensitive to oligosaccharides [48,49]. In addition, FACE has several other advantages, including its simple handling and low experimental costs. FACE is, however, not able to analyze the structure of the individual chitooligosaccharides; the molecular composition of the individual products will be the focus of our further research that will employ nuclear magnetic resonance as well as mass spectrometry analysis.

Different chemical or enzymatic treatments can carry out the production of chitooligosaccharides from chitin and chitosan. Enzymatic preparations have attracted a lot of attention due to their simplicity and environmental friendliness because they do not require harsh thermochemical processes, strong physical forces, or harmful chemicals [50,51]. This report provides insights into novel enzymatic methods for producing chitooligosaccharides, with potential applications in various fields.

Various Chia has been reported in mammals and birds, while mouse Chia has been studied the most extensively so far [8,21,22,23,24,52]. Recently, Du et al. reported an increased yield of (GlcNAc)_2_ by recombinant mouse Chia produced in *Pichia pastris* using industrial and agricultural applications [53]. Compared to mouse enzymes, monkey CHIA has three advantages. First, monkey CHIA is a highly active chitinase under a broad pH range [44]. Second, this enzyme efficiently degrades chitin into (GlcNAc)_2_ and is stable at high temperatures (Figure 1). Finally, monkey CHIA degrades chitin and chitosan substrates, producing not only dimers but also longer chitooligosaccharides ((GlcNAc)_2–6_), respectively (Figure 2 and Figure 3).

It was reported that chitin oligosaccharides with lower molecular weights ((GlcNAc)_2–4_) induced genes related to vegetative growth, development, and carbon and nitrogen metabolism in *Arabidopsis thaliana* [30]. Moreover, (GlcNAc)_6–8_ chitooligosaccharides promoted the growth of wheat seedlings under salt stress [29]. Thus, monkey CHIA can be used to produce the low-to-moderate length chitooligosaccharides, promoting plant growth for agricultural applications. We showed that chitosan degradation catalyzed by monkey CHIA produces chitooligosaccharides longer than hexamers (Figure 3). Although the molecular mechanism of the biological activity of chitooligosaccharides is not fully understood, the monkey CHIA described here may offer new options for the preparation of chitooligosaccharides applicable in numerous sectors.

Chitooligosaccharides also possess bioactivities such as anti-microbial, anti-tumor, and anti-inflammatory effects and drug delivery [25,26,27,28,29]. Chitooligosaccharides with a degree of polymerization of less than 10 are typically water-soluble, and it makes them an extremely valuable product in a wide range of industries. Porcine and mouse Chia recognize GlcNAc residues of chitosan and produce hetero-chitooligosaccharides [37,38].

CHIA is involved in the pathogenesis of asthma and pulmonary fibrosis [20]. Studies in Chia-deficient mice have shown that chitin polymers accumulate in the airways, causing pulmonary fibrosis that can be ameliorated by administering active enzymes [20]. In humans, CHIA’s expression levels [54] and chitin-degrading activity [19] are very low. Several studies on the activation of the human enzyme have been reported using single nucleotide polymorphisms [19] or mutagenesis achieved by an error-prone PCR system [33]. Increasing chitinase activity in humans could be a preventive and/or therapeutic measurement for lung disease. Due to its high homology with human CHIA, monkey CHIA might potentially be adjusted for future purposes. For example, microbial chitinases have been used in a wide range of applications [55]. CHIA could therefore become a robust tool in chitooligosaccharide production for agricultural and biomedical purposes (Figure 5).

## 4. Materials and Methods

### 4.1. Preparation of the Recombinant Monkey CHIA Expressed in E. coli

As described previously, we expressed monkey CHIA as a fusion protein of Protein A-monkey CHIA-V5-His in *E. coli* [44]. We purified monkey CHIA using an IgG Sepharose column (G.E. Healthcare, Piscataway, NJ, USA) described previously [44]. The protein fractions were separated using standard SDS-PAGE, followed by Western blot using a polyclonal anti-V5-HRP monoclonal antibody (Invitrogen, Carlsbad, CA, USA). The proteins were quantified by a Luminescent Image Analyzer (Amersham ImageQuant 800, G.E. Healthcare, Piscataway, NJ, USA). The gel was then stained by SYPRO Ruby (Thermo Fisher Scientific, Waltham, MA, USA) and quantified again by the Luminescent Image Analyzer. Protein concentration was determined by the Protein Assay (Bio-Rad, Richmond, CA, USA) based on the method of Bradford [56], with bovine serum albumin as the standard.

### 4.2. Chitin and Chitosan Substrates

We used three types of chitins (α-chitin, P-chitin, and colloidal chitin) and two types of chitosan (random-type chitosan and block-type chitosan) as substrates. Shrimp shell α-chitin was purchased from Sigma-Aldrich, St. Louis, MO, USA. P-chitin was a polymeric form of chitin (P-CHITN, Megazyme, Bray, Ireland). Colloidal chitin was prepared from α-chitin, as described previously [45]. Random-type chitosan (Viscosan, FLEXICHEM, Uttran, Sweden) is a homogenously deacetylated chitin (D.D. 70%). Heterogeneously deacetylated chitosan (block-type chitosan) with D.D. 69% was a generous gift from Funakoshi Co., Ltd., Tokyo, Japan.

### 4.3. Degradation of Chitin and Chitosan by Recombinant Monkey CHIA

Chitin or chitosan substrates (1 mg/reaction) were incubated with recombinant monkey CHIA at pH 2.0 or pH 5.0 in McIlvaine’s buffer (0.1 M citric acid and 0.2 M Na_2_HPO_4_) at 50 °C for 1, 3, 5, 24, 48, or 72 h.

### 4.4. Analysis of Chitooligosaccharides by FACE

The degradation products were labeled and separated by fluorophore-assisted carbohydrate electrophoresis (FACE), as described previously [22,23,38,42]. The samples were quantified using a Luminescent Image Analyzer.

## 5. Conclusions

Crab-eating monkey CHIA has robust chitinolytic activity and is stable under strong acid and high-temperature conditions. We showed that this enzyme efficiently degrades chitin and chitosan at high temperatures (50 °C). Superior substrates for monkey CHIA for the GlcNAc dimer and chitooligosaccharides are P-chitin and random-type chitosan, respectively. Thus, crab-eating monkey CHIA has a promising potential to produce different chitooligosaccharides for agricultural and biomedical purposes.

## Figures and Tables

**Figure 1 molecules-27-00409-f001:**
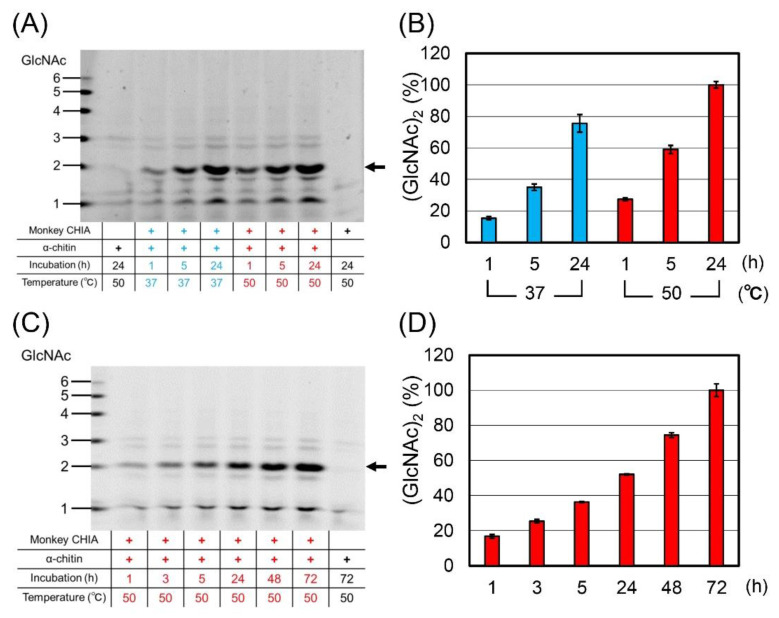
Degradation of chitin by monkey CHIA at high temperatures. (**A**) Degradation of α-chitin at 37 °C or 50 °C and pH 5.0. Size standards on the left margin are defined as chitin oligomers. (**B**) Quantitative data of (GlcNAc)2 at each temperature. Fluorescence intensity is estimated from the results in (**A**). The quantitative data express the percentage of the maximum signal of (GlcNAc)_2_ achieved at 24 h and 50 °C that was set to 100%. (**C**) The time-dependent change in degradation products from α-chitin at 50 °C. Size standards on the left margin are defined as chitin oligomer standards. (**D**) Quantitative data of (GlcNAc)_2_ for each incubation time. Fluorescence intensity is estimated from the results in (**C**). The quantitative data express the percentage of the maximum signal of (GlcNAc)_2_ achieved at 72 h, set to 100%. The images in (**A**,**C**) were cropped from red dotted lines on original full-length gel images, shown in Appendix A. Error bars represent mean ± standard deviation from a single experiment conducted in triplicate (Appendix A).

**Figure 2 molecules-27-00409-f002:**
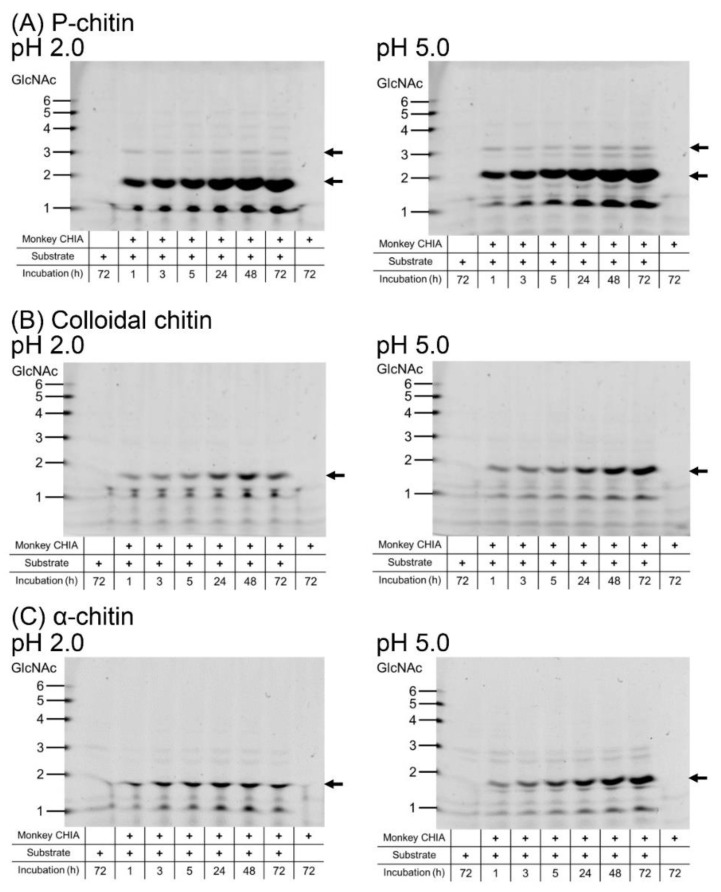
Fluorophore-assisted carbohydrate electrophoresis (FACE) analysis of chitin substrate degradation by monkey CHIA. The substrates, P-chitin (**A**), colloidal chitin (**B**), and α-chitin (**C**), were incubated with monkey CHIA at pH 2.0 or 5.0 and 50 °C for 1, 3, 5, 24, 48, and 72 h. Left panels show results of the reactions at pH 2.0; right panels show results of the reactions at pH 5.0. Full-length gel images are shown in Appendix A. The resulting products were analyzed by the FACE method. Size standards on the left margin are defined as chitin oligomers.

**Figure 3 molecules-27-00409-f003:**
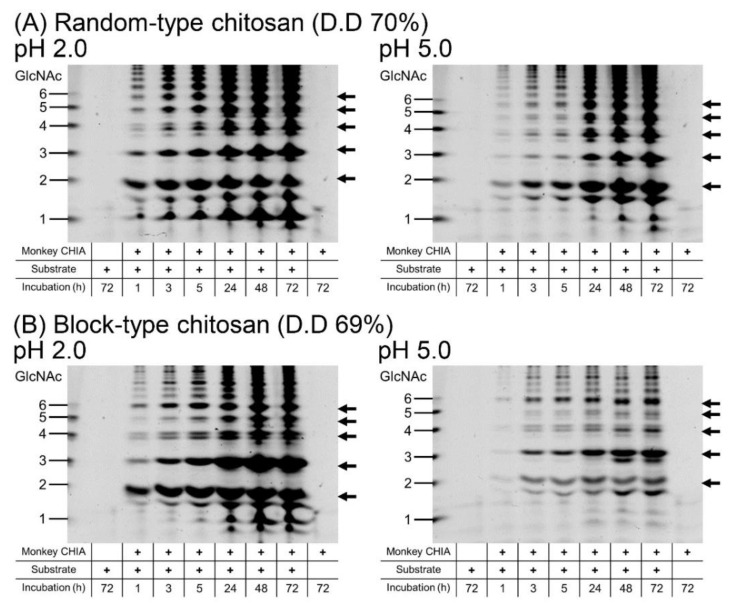
FACE analysis of chitosan substrate degradation by monkey CHIA. The substrates, random-type (**A**) and block-type chitosan (**B**) were incubated with monkey CHIA at pH 2.0 or 5.0 and 50 °C for 1, 3, 5, 24, 48, and 72 h. Left panels show results of the reactions at pH 2.0; right panels show results of the reactions at pH 5.0. Full-length gel images are shown in Appendix A. Size standards on the left margin are defined as chitin oligomers.

**Figure 4 molecules-27-00409-f004:**
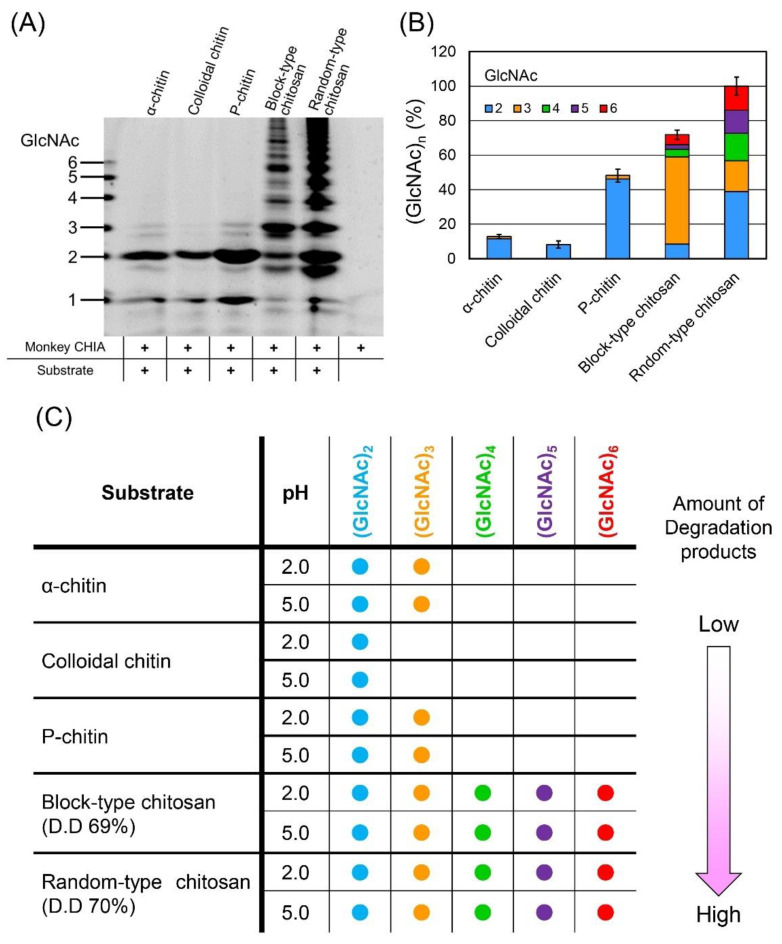
Comparison of degradation patterns of chitin and chitosan substrates. Five substrates were incubated with monkey CHIA at pH 5.0 and 50 °C for 72 h. Those included α-chitin, colloidal chitin, P-chitin, and block-type and random-type chitosan. (**A**) FACE analysis of chitin and chitosan degradation by monkey CHIA. Full-length gel images are shown in Appendix A. Size standards on the left margin are defined as chitin oligomers. (**B**) Quantitative data of (GlcNAc)_2–5_ were obtained from each substrate. The results are shown in Appendix A. The quantitative data that express the percentage of the maximum signal of degradation products (the total amount of degradation products from random-type chitosan) was set to 100%. Error bars represent mean ± standard deviations from a single experiment conducted in triplicate (Appendix A). (**C**) Qualitative analysis of the produced chitooligosaccharides from each substrate.

**Figure 5 molecules-27-00409-f005:**
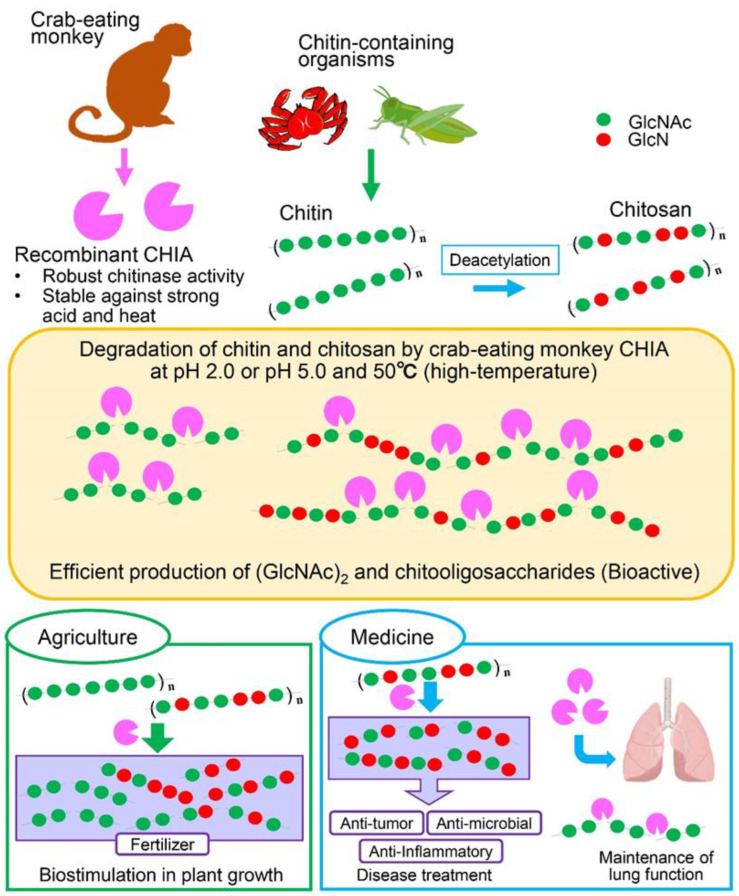
Efficient chitin and chitosan degradation by monkey CHIA at a high temperature (50 °C) produces different chitooligosaccharides. The higher temperature increased the efficiency of substrate degradation, with chitin and chitosan being processed to (GlcNAc)_2_ and chitooligosaccharides with variable lengths, respectively. The crab-eating monkey CHIA has a promising potential for producing chitooligosaccharides for agricultural and biomedical purposes.

## Data Availability

Data supporting the reported results are available from the corresponding author (Fumitaka Oyama).

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
