# Peer review of "Crab-Eating Monkey Acidic Chitinase (CHIA) Efficiently Degrades Chitin and Chitosan under Acidic and High-Temperature Conditions"

_molecules, 2022, doi:10.3390/molecules27020409_

Round 1

Reviewer 1 Report

The manuscript describes a novel CHIA  from crab eating monkey and utilized it for for production of chito-oligosachharides. The study is nicely planned and executed. 

The reason for production of different proportion of chitooligomers in figure 4-B may be elaborated in discussion. Can the incubation conditions be modified so that only particular oligomer is produced. Discuss

Figure 5 is not necessary as the message regarding different usage of CHIA has been conveyed in text very well. Cite some recent review indicating applications of the chitinases like Biology 202110(12), 1319

Author Response

Point-by-Point Replies to Reviewers

Our responses to the reviewers in this letter and the revised main text are written in corresponding colors.

Red, Reviewer #1; Blue, Reviewer #2

Reviewer #1 's Comments:

Comment 1: <The manuscript describes a novel CHIA from crab eating monkey and utilized it for for production of chito-oligosachharides. The study is nicely planned and executed.

The reason for production of different proportion of chitooligomers in figure 4-B may be elaborated in discussion. Can the incubation conditions be modified so that only particular oligomer is produced. Discuss>

Response to Reviewer #1 Comment 1:

Thank you for your comments.

We discuss the different proportions of chitooligosaccharides in Figure 4B. We suggest that the substrates structure cause the difference in produced chitooligosaccharides. We added the description on these results (Figure 4B) in Results (lines 193-194) and discuss them in the Discussion (231-237) in the revised version.

In accordance with your comment, we also discuss the production of particular chitooligosaccharides. We suggest that specific chitooligosaccharides could be obtained by a short-time incubation at suitable pH . As shown in Figure 3, a specific chitooligosaccharide was obtained as the main product in the short-term reaction (~ 5 h). In addition, GlcNAc dimer and longer chitooligosaccharides were produced in large quantities and their composition depended on the substrate type and the pH condition. Thus, it is possible to control a particular amount of chitooligosaccharides by selecting the substrate, pH and reaction time. These were included in the Discussion (lines 216-222).

Comment 2: <Figure 5 is not necessary as the message regarding different usage of CHIA has been conveyed in text very well. Cite some recent review indicating applications of the chitinases like Biology 2021, 10(12), 1319>

Response to Reviewer #1 Comment 2:

We think Figure 5 would help the readers to understand the present study comprehensively. Therefore, we would like to include this Figure in our manuscript.

We cited the suggested recent review as Reference #55 and modified the description in the Discussion (lines 290-293).

Reviewer 2 Report

Chitooligosaccharides have remarkable anti-microbial, anti-tumor, and anti-inflammatory bioactivities. Enzymatic production of chitooligosaccharides by chitosanases or chitinases is preferred. Therefore, finding chitosanases or chitinase with novel properties is required. In this study, a crab-eating monkey acidic chitinase (CHIA) was successfully overexpressed in E. coli, and was used to degrade five different substrates under different conditions. Some interesting results were obtained. However, in opinion of the reviewer, some issues need to be addressed prior to publication. Please check the below.

  1. Western blot analysis of the recombinant proteins was provided in supporting materials (Figure S1). However, SDS PAGE analysis and the yield of the recombinant proteins were not shown, which are important for overexpression of recombinant proteins too. Please provide them.
  2. According to FACE analysis of degradation products of chitin by CHIA, it seems that GlcNAc was also generated besides (GlcNAc)2, and no other intermediates were observed. Could author clarify that CHIA belongs to exo or endo chitinases and which GH family it belongs to?
  3. In Figure 1A, 1C, and Figure 2, could authors clarify what the weak bands between the (GlcNAc)n standards are?
  4. It was interesting to see that CHIA behaved differently against different types of chitin and chitosan substrates. Could authors explain why??
  5. Based on FACE analysis of degradation products of chitosan by CHIA (Figure 3 and Figure 4A), the size distribution of produced chitooligosaccharides was shown. However, it is not clear what the components (numbers of GlcNAc and GlcN) of chitooligosaccharides were. Therefore, MALDI-TOF-MS analysis of degradation products of chitosan is needed.

Author Response

Point-by-Point Replies to Reviewers

Our responses to the reviewers in this letter and the revised main text are written in corresponding colors.

Red, Reviewer #1; Blue, Reviewer #2

Reviewer #2 's comments:

Comment 1: <Chitooligosaccharides have remarkable anti-microbial, anti-tumor, and anti-inflammatory bioactivities. Enzymatic production of chitooligosaccharides by chitosanases or chitinases is preferred. Therefore, finding chitosanases or chitinase with novel properties is required. In this study, a crab-eating monkey acidic chitinase (CHIA) was successfully overexpressed in E. coli, and was used to degrade five different substrates under different conditions. Some interesting results were obtained. However, in opinion of the reviewer, some issues need to be addressed prior to publication. Please check the below.

Western blot analysis of the recombinant proteins was provided in supporting materials (Figure S1). However, SDS PAGE analysis and the yield of the recombinant proteins were not shown, which are important for overexpression of recombinant proteins too. Please provide them.>

Response to Reviewer #2 Comment 1:

Thank you for your comments and suggestions.

According to your suggestion, we included SDS-PAGE gel stained by SYPRO Ruby (Supplementary Figure S1C) and the yield of the recombinant proteins. These are described in the Results (lines 99-104) and Materials and Methods section (lines 304-311) in the revised version.

Comment 2: <According to FACE analysis of degradation products of chitin by CHIA, it seems that GlcNAc was also generated besides (GlcNAc)2, and no other intermediates were observed. Could author clarify that CHIA belongs to exit or endo chitinases and which GH family it belongs to?>

Response to Reviewer #2 Comment 2:

CHIA belongs to the GH18 family. At present, its chitinolytic properties are considered as endo chitinases.

We described these in the Introduction (lines 47-51).

Comment 3: <In Figure 1A, 1C, and Figure 2, could authors clarify what the weak bands between the (GlcNAc)n standards are?>

Response to Reviewer #2 Comment 3:

The weak bands below the main product (Figures 1-3) are hetero chitooligosaccharides containing GlcNAc and GlcN.

These are included in the Discussion section (lines 238-244).

Comment 4: <It was interesting to see that CHIA behaved differently against different types of chitin and chitosan substrates. Could authors explain why??>

Response to Reviewer #2 Comment 4:

According to your and Reviewer #1’s comments, we discussed the differences in chitooligosaccharides production. We suggest that the substrates structure cause the difference in produced chitooligosaccharides. First, the degree of deacetylation influences the pattern and amount of the degradation. We have previously reported [37, 38] that degradation products from β-chitin having higher degree of deacetylation (DD)) were more abundant in comparison with α-chitin.

The block-type chitosan has amorphous regions, which are highly deacetylated areas (GlcN-rich). On the other hand, the GlcNAc and GlcN residues are interspersed in the random type molecules. As shown in this study, monkey CHIA preferentially degrades the randomly placed GlcNAc-rich regions producing more variable-sized chitooligosaccharides and influencing the amount of degradation products.

We described in the Discussion (lines 228-237).

Comment 5: <Based on FACE analysis of degradation products of chitosan by CHIA (Figure 3 and Figure 4A), the size distribution of produced chitooligosaccharides was shown. However, it is not clear what the components (numbers of GlcNAc and GlcN) of chitooligosaccharides were. Therefore, MALDI-TOF-MS analysis of degradation products of chitosan is needed.>

Response to Reviewer #2 Comment 5:

This study evaluated chitooligosaccharides produced from chitin and chitosan using FACE because this method is very sensitive for oligosaccharides detection. In addition, FACE has several advantages including simple handling and low experimental costs.

As you mentioned, the exact composition of the chitooligosaccharides in terms of GlcNAc and GlcN content is unclear. FACE is not able to analyze the structure of the chitooligosaccharides and the molecular composition of the individual products is the focus of our further research that will employ NMR and MS analysis.

We included this description in Discussion (lines 245-250).